# Clinical features of talaromycosis in people living with HIV/AIDS (PWHA) and patients with anti-interferon-γ autoantibodies

Kawisara Krasaewes[1], Narootchai Patanadamrongchai[2], Quanhathai Kaewpoowat[1,3], Jiraprapa Wipasa[4], Saowaluck Yasri[1], Antika Wongthanee[2], Romanee Chaiwarith[1]*

1 Division of Infectious Diseases and Tropical Medicine, Department of Internal Medicine, Faculty of Medicine Chiang Mai University, Chiang Mai, Thailand, 2 Department of Internal Medicine, Faculty of Medicine Chiang Mai University, Chiang Mai, Thailand, 3 Department of Internal Medicine, University of Iowa, Iowa City, Iowa, United States of America, 4 Research Institute for Health Sciences, Chiang Mai University, Chiang Mai, Thailand

* rchaiwar@gmail.com

## Abstract

### Background

Talaromycosis is increasingly reported in HIV-uninfected, immunocompromised individuals in an endemic area. The aim of this study was to compare the characteristics and mortality associated with talaromycosis in adult immunocompromised individuals caused by the anti-interferon-gamma autoantibody (anti-IFN- γ AAb) with those of people living with HIV/AIDS (PWHA).

### Methods

A retrospective cohort study was conducted at Maharaj Nakorn Chiang Mai Hospital, Thailand, in adults with confirmed HIV infection or anti-IFN- γ-AAbs diagnosed with talaromycosis.

### Results

Thirty-two patients with anti-IFN- γ-AAbs and 235 PWHA were included. Patients with anti-IFN- γ-AAbs were older and more likely to have comorbidities. PWHA were more likely to have constitutional symptoms, cough, dyspnea, diarrhea, splenomegaly, umbilicated skin lesions, abnormal chest radiographs, and fungemia. Patients with anti-IFN- γ-AAbs were more likely to have skin lesions such as macule/papules/nodules, abscesses and Sweet's syndrome, as well as bone and joint infections and higher white blood cell counts. The time from first symptom to treatment was longer in patients with anti-IFN- γ-AAbs (44.5 days vs. 30.0 days, p-value = 0.049). The 24-week mortality rate was 9.4% (3 patients) in patients with anti-IFN- γ-AAbs and 15.3% (36 patients) in PWHA (p-value = 0.372).

which permits unrestricted use, distribution, and reproduction in any medium, provided the original author and source are credited.

**Data availability statement:** A publicly available dataset, representing 25% of the total data and excluding potentially identifying or sensitive patient information, is accessible at http://datadryad.org/stash/share/6ZhPOCnttzu1dB5_dSPHjH2re1duibabSXtB3YOIXPI. The remaining datasets are available from the corresponding author upon reasonable request.

**Funding:** The author(s) received no specific funding for this work.

**Competing interests:** The authors have declared that no competing interests exist.

## Conclusions

The clinical features of talaromycosis in patients with anti-IFN- γ-AAbs differed from PWHA. Clinicians in areas where talaromycosis is endemic should be aware of the different features of talaromycosis in patients with anti-IFN- γ AAbs.

## Author summary

Talaromycosis, caused by *Talaromyces marneffei*, is an opportunistic fungal infection in people with a weakened immune system. The number of cases increased immediately after the HIV epidemic. HIV infection weakens the immune system by destroying CD4+T cells, which are among the most important components of the immune system to protect against *T. marneffei*. However, with the introduction of antiretroviral therapy, the number of cases in people with HIV/AIDS (PWHA) has declined. Recently, we have found a group of patients whose immune system is weakened by autoantibodies against interferon-gamma. In these patients, CD4+T cells are normal, but the cytokine cascade cannot fight the fungus due to antibodies against interferon-gamma (anti-IFN- γ AAbs), a major cytokine in the proinflammatory cytokine cascade. In this study, the clinical features of talaromycosis were compared between PWHA and patients with anti-IFN- γ AAbs. We found that patients with anti-IFN- γ AAbs had more diverse types of skin and subcutaneous lesions including macule, papules, nodules, abscesses, and Sweet's syndrome, as well as bone and joint infections, while PWHA were more likely to have constitutional symptoms, cough, dyspnea, diarrhea, splenomegaly, and umbilicated skin papules. Our paper highlights the different clinical characteristics of talaromycosis in these two populations.

## Introduction

Talaromycosis, a disseminated infection caused by *Talaromyces marneffei*, is a common opportunistic endemic fungal infection in people with HIV/AIDS (PWHA) living in tropical and subtropical Asia, especially when the CD4 cell count is < 100 cells/μL. [1]. The clinical features of talaromycosis in PWHA are well described in the literature, with classic central necrotic (umbilicated) papules appearing on the face, trunk and extremities, fever, weight loss, hepatosplenomegaly, lymphadenopathy, respiratory and gastrointestinal abnormalities [1–3]. The incidence of talatomycosis in PWHA has decreased dramatically since the introduction of antiretroviral therapy [4]. However, talaromycosis has been increasingly observed in immunocompromised non-HIV hosts with an impaired cell-mediated immune response, such as those with autoimmune diseases, cancer and organ transplants, as well as those receiving immunosuppressive therapy [4]. Recently, there has been a growing number of studies of adult-onset immunodeficiency (AOID) caused by anti-interferon-γ autoantibodies (anti-IFN- γ AAbs) [5–8]. This condition is reported to be most prevalent in China and Southeast Asia,

including Thailand, an area where *T. marneffei* is endemic. Talaromycosis has been reported to occur in these patients alongside other disseminated infections with tuberculous mycobacteria, non-tuberculous mycobacteria and other dimorphic fungal infections [5–8]. Data on the clinical features of talaromycosis in patients with anti-IFN- γ AAbs are limited [7,9]. We previously reported the clinical features of patients with talaromycosis in HIV-uninfected patients between 2007 and 2011; however, the cause of immunodeficiency was not fully investigated [10]. Here, we described clinical features and outcomes of talaromycosis in patients with confirmed anti-IFN- γ AAbs and PWHA in northern Thailand, where *T. marneffei* is endemic.

## Methods

### Ethics statement

The study was approved by the Research Ethics Committee, Number 4, of the Faculty of Medicine, Chiang Mai University. The approval certificate number is 224/2021. Due to the retrospective nature of the study, written/verbal informed consent was not obtained.

A retrospective cohort study was conducted at Maharaj Nakorn Chiang Mai Hospital, a teaching hospital of Chiang Mai University, Chiang Mai, Thailand. Patients aged 18 years and older with the ICD-10 code B48.4 (talaromycosis) attending Maharaj Nakorn Chiang Mai Hospital were studied. Those who met the definition of talaromycosis, defined by a positive culture of *T. marneffei* from sterile sites or proven by histopathology, with documentation of HIV infection according to the national standard guideline [11] or detection of anti-IFN- γ AAbs by ELISA [8], were included in the study. During the study period from January 1, 2012 to June 30, 2021, 32 patients with anti-IFN- γ AAbs and 235 PWHA diagnosed with talaromycosis were included in the analysis. The flow chart of the participants is shown in Fig 1.

Demographic data, clinical characteristics, laboratory findings, treatment and treatment outcomes were reviewed and extracted from the medical and laboratory records. If a patient lost to follow up or transferred to another hospital, the patient's status was obtained from the Thai civil registry. In the case of a deceased patient, the date of death was determined.

### Laboratory test for antibody to IFN-γ using ELISA

The test for antibodies to IFN-γ was performed using an enzyme-linked immunosorbent assay (ELISA), which was modified from the methods previously described by Tang et al [12]. In brief, 2% skimmed milk in phosphate-buffer saline (PBS) was used in blocking buffer and diluent, replacing the 5% normal goat serum used in the original method. Additionally, o-phenylenediamine dihydrochloride (OPD) was utilized as the substrate instead of tetramethylbenzidine (TMB), as described by Tang et al [12].

### Opportunistic infection definitions

Concurrent opportunistic infections (OI) and previous OI are OI diagnosed within 2 weeks and more than 2 weeks after the diagnosis of talaromycosis, respectively.

### Data analysis

Data were expressed as number and persentage for categorial data, or mean and standard deviation (SD) or median (interquartile range; IQR) for continuous data. Comparisons between groups were performed using the chi-square test or Fisher's exact test for categorical data and Student's t-test or Mann-Whitney U test for continuous data. The cumulative probability of death was calculated and presented with Kaplan-Meier curves. The difference in mortality rate between 2 groups was compared using the log-rank test. Statistical analyses were performed with R software Ver.4.1.2. A two-sided test at a p-value of < 0.05 was used to indicate statistical significance.

A publicly available dataset, representing 25% of the total data and excluding potentially identifying or sensitive patient information, is accessible at http://datadryad.org/stash/share/6ZhPOCnttzu1dB5_dSPHjH2re1duibabSXtB3YOIXPI [13]. The remaining datasets are available from the corresponding author upon reasonable request.

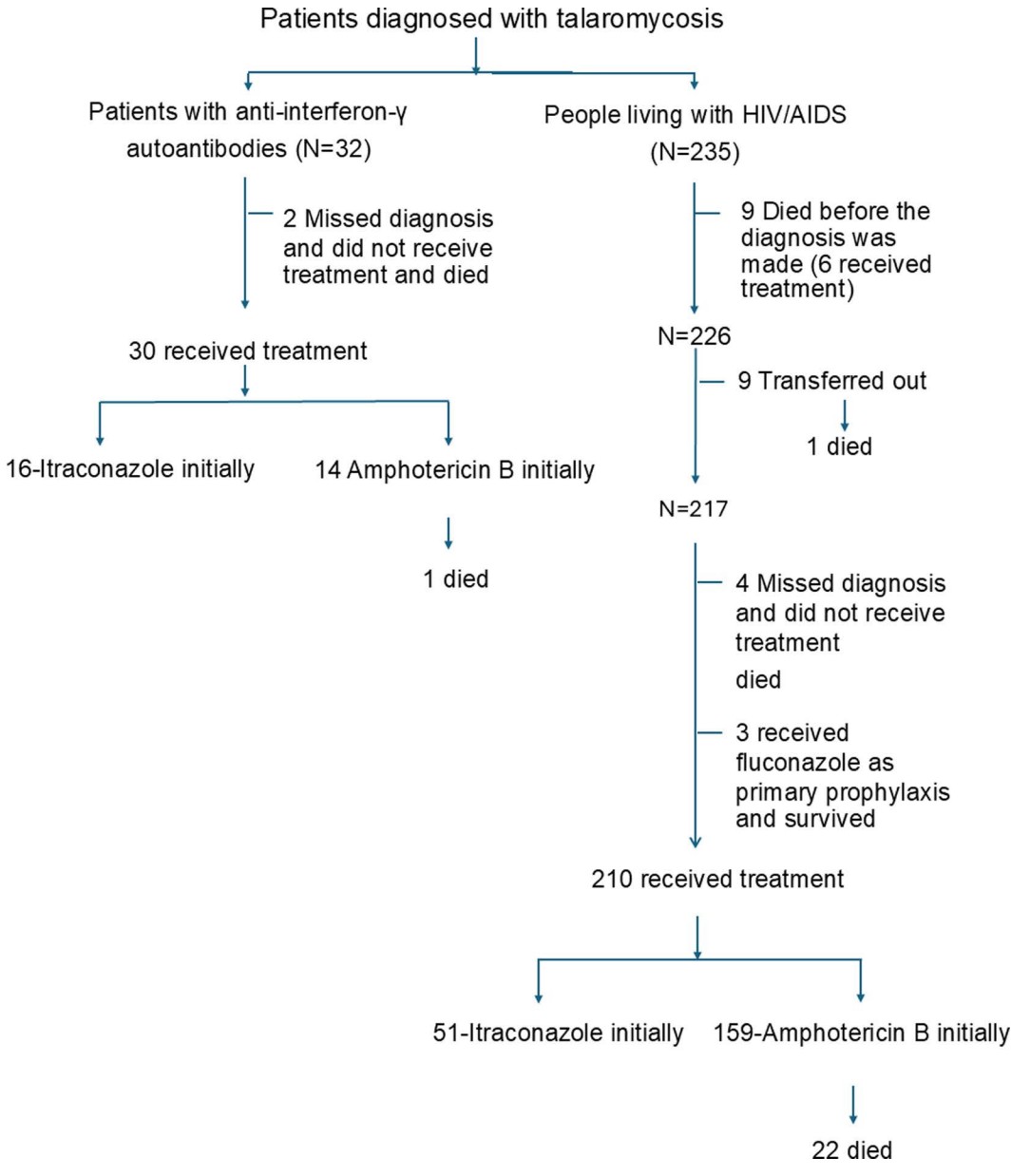

**Fig 1. Flow chart, treatment, and outcome of the study participants.**

## Results

Thirty-two patients with anti-IFN- γ-AAbs had a median (IQR) age of 57.5 (52.8, 63.0) years, and 19 of them (59.4%) were male. Nineteen patients had comorbidities (59.4%). Two hundred and thirty-five PWHA had a median (IQR) age of 35.0 (29.0, 43.0) years, and 174 of them (74.0%) were male. Patients with anti-IFN- γ-AAbs were more likely to be older and to have underlying diseases, particularly hypertension, dyslipidemia, and chronic kidney disease. (Table 1)

**Table 1. Demographic and clinical characteristics of patients with talaromycosis between patients with anti-IFN- γ AAbs and people with HIV/AIDS (PWHA).**

| Demographic characteristics | Patients with anti-IFN- γ AAbs (n = 32) | PWHA (n = 235) | p-value |
|---|---|---|---|
| Male | 19 (59.4) | 174 (74.0) | 0.082 |
| Age (years) | 57.5 (52.8, 63.0) | 35.0 (29.0, 43.0) | <0.001 |
| Co-morbidities | 19 (59.4) | 49 (20.9) | <0.001 |
| Hypertension | 8 (25) | 7 (3.0) | <0.001 |
| Diabetes mellitus | 2 (6.3) | 4 (1.7) | 0.154 |
| Chronic kidney diseases | 4 (12.5) | 1 (0.4) | <0.001 |
| Dyslipidemia | 3 (9.4) | 4 (1.7) | 0.039 |
| Cirrhosis | 1 (3.1) | 3 (1.3) | 0.402 |
| Malignancy | 1 (3.1) | 2 (0.9) | 0.319 |
| Thalassemia | 1 (3.1) | 3 (1.3) | 0.419 |
| Talaromycosis was the first OI | 11 (34.4) | 192 (81.7) | <0.001 |
| Previous/ concurrent OIs* | | | |
| Tuberculosis | 10 (31.3) | 30 (12.7) | 0.014 |
| Cryptococcosis | 0 | 16 (6.8) | 0.231 |
| Non-tuberculous mycobacteria | 20 (62.5) | 4 (1.7) | <0.001 |
| Salmonellosis | 7 (21.9) | 18 (7.7) | 0.018 |
| Oropharyngeal and esophageal candidiasis | 0 | 94 (40.0) | <0.001 |
| Cytomegalovirus infection | 1 (3.1) | 7 (3.0) | 1.000 |
| Pneumocystis pneumonia | 1 (3.1) | 47 (20.0) | 0.015 |
| Cerebral toxoplasmosis | 0 | 3 (1.3) | 1.000 |
| **Clinical characteristics** | | | |
| Median (IQR) days from first symptom to diagnosis | 30.0 (14.0, 82.5) | 30.0 (17.5, 60.0) | 0.643 |
| **Signs/symptoms** | | | |
| Fever | 21 (65.6) | 208 (88.5) | 0.002 |
| Fatigue | 1 (3.1) | 114 (48.5) | <0.001 |
| Weight loss | 8 (25.0) | 129 (54.9) | 0.002 |
| Anorexia | 5 (15.6) | 110 (46.8) | <0.001 |
| Cough | 6 (18.7) | 105 (44.7) | 0.005 |
| Dyspnea | 3 (9.4) | 59 (25.1) | 0.048 |
| Diarrhea | 2 (6.3) | 53 (22.6) | 0.032 |
| Abdominal pain | 1 (3.1) | 36 (15.3) | 0.096 |
| Hepatomegaly | 7 (21.9) | 63 (26.8) | 0.552 |
| Splenomegaly | 0 | 49 (20.9) | 0.004 |
| Skin lesions | 22 (68.8) | 123 (52.3) | 0.080 |
| Macule/patch | 4 (12.5) | 7 (3.0) | 0.031 |
| Papule/plaque | 6 (18.7) | 17 (7.2) | 0.042 |
| Pustule | 1 (3.1) | 0 | 0.120 |
| Vesicle/bullae | 1 (3.1) | 0 | 0.120 |
| Nodule | 4 (12.5) | 1 (0.4) | <0.001 |
| Abscess | 9 (28.1) | 0 | <0.001 |
| Ulcer | 2 (6.3) | 2 (0.9) | 0.072 |
| Umbilicated lesion | 4 (12.5) | 104 (44.3) | <0.001 |
| Sweet's syndrome | 2 (6.3) | 0 | 0.014 |
| Lymphadenopathy | 20 (62.5) | 99 (42.1) | 0.030 |

*(Continued)*

**Table 1.** (Continued)

| Demographic characteristics | Patients with anti-IFN- γ AAbs (n = 32) | PWHA (n = 235) | p-value |
|---|---|---|---|
| Bone and joint infections** | 12 (37.5) | 5 (2.1) | <0.001 |

*1 patient might have > 1 OI

**Signs/symptoms of bone and joint infections include arthralgia, arthritis, osteomyelitis, spondylodiscitis, paravertebral abscess

Abbreviations: OI, opportunistic infection

Data are presented in number (%), mean (±standard deviation) or median (interquartile range)

## Clinical characteristics

Compared to PWHA, patients with anti-IFN- γ AAbs had less frequent constitutional symptoms (fever, fatigue, weight loss, anorexia), cough, dyspnea, diarrhea, splenomegaly, and umbilicated skin lesions, but more frequent bone and joint involvement such as arthritis, osteomyelitis, spondylodiscitis, and paravertebral abscesses. The skin lesions commonly seen in patients with anti-IFN- γ AAbs were more diverse and include macules/patch, papules/plaques, nodules, abscesses and Sweet's syndrome whereas PWHA were more likely to have central umbilicated skin lesions. (Table 1)

## Laboratory and findings

Patients with anti-IFN- γ-AAbs more frequently had higher white blood cells, eosinophil, platelet counts and alkaline phosphatase level, less frequently transaminitis and fungemia than PWHA. Three PWHA had neither a CD4 count < 200 cells/μL nor a CD4 < 14%. The CD4 counts (%) of these 3 PWHA at the time of diagnosis of talaromycosis were 264 cells/μL (18%), 325 cells/μL (20%) and 527 cells/μL (23%) (30%). These 3 patients had no other immunosuppressive conditions. Patients with anti-IFN- γ AAbs were less likely to have an abnormal chest radiograph. Alveolar infiltration and pleural effusion were more common in patients with anti-IFN- γ AAbs, whereas interstitial infiltration was less common (Table 2).

## Treatment and outcomes

Two patients with anti-IFN-γ autoantibodies (AAbs) were not diagnosed with talaromycosis, were not treated, and subsequently died. Among the 30 patients who received treatment, 29 (96.7%) were initially treated with itraconazole, and all survived. One patient, who was treated with amphotericin B followed by itraconazole, died of hospital-acquired pneumonia despite showing clinical improvement after treatment for talaromycosis.

Nine PWHA died before a diagnosis of talaromycosis was made; six of them received empirical treatment with amphotericin B, while three did not received any treatment. Among the remainder 217 patients, nine were transferred to another hospital, including one who died. Four patients missed the diagnosis of talaromycosis, were not treated, and died. Three patients missed the diagnosis of talaromycosis but received fluconazole as primary prophylaxis and survived. Of 216 patients who received treatment, 51 (53.3%) were initially treated with itraconazole and all survived. Ninety-seven patients were initially treated with amphotericin B, 22 of whom died. (Fig 1)

The time from first symptom to treatment and the time from diagnosis to treatment was longer in patients with anti-IFN-γ-AAbs. (Table 2).

**Mortality.** Three patients with anti-IFN- γ AAbs and 36 PWHA died with a 24-week mortality of 9.4%% and 15.3%, respectively. The median time from diagnosis to death was 30 days (IQR 21.5-60.0) for patients with anti-IFN- γ AAbs and 7 days (IQR 0.8-35.0) for PWHA (p-value = 0.111). The probability of death is shown in Fig 2. The causes of death were attributed to talaromycosis in 18 patients, cryptococcosis in 1 patient, disseminated tuberculosis in 1 patient, salmonellosis in 1 patient, *Pneumocystis* pneumonia in 1 patient, and cytomegalovirus pneumonia in 1 patient. Additionally, 9 patients

**PLOS Neglected Tropical Diseases**

**Table 2. Laboratory findings and treatment outcome of patients with talaromycosis in patients with anti-IFN- γ AAbs and people with HIV/AIDS (PWHA).**

| Variables | Patients with anti-IFN- γ AAbs (n = 32) | PWHA (n = 235) | p-value |
|---|---|---|---|
| **Laboratory findings** | | | |
| Hemoglobin (g/dL), mean±SD | 9.0 ± 1.7 | 9.6 ± 2.3 | 0.205 |
| White blood cells (x1000/μL) median (IQR) | 18.1 (13.6, 23.8) | 4.2 (3.2, 6.0) | <0.001 |
| % Eosinophil | 3.8 (2.7, 6.2) | 0.7 (0.2, 1.9) | <0.001 |
| Platelets (x1000/μL) | 479.0 (314.0, 548.0) | 149.0 (80.5, 253.3) | <0.001 |
| Albumin (g/dL) | 2.6 ± 0.7 | 2.9 ± 0.8 | 0.111 |
| Alanine aminotransferase (U/L) | 19 (12, 29) | 30 (21, 48) | 0.001 |
| Alkaline phosphatase (U/L) | 203.0 (139.0, 343.0) | 134.5 (80.5, 280.5) | 0.012 |
| CD4 cells (cells/μL) | 804.0 (489.0, 1,189.0) (N = 9) | 18.0 (8.0, 40.0) (N = 223) | <0.001 |
| Fungemia | 12/26 (46.2) | 190/226 (84.1) | <0.001 |
| **Chest radiograph** | N = 28 | N = 223 | |
| Abnormal | 9 (32.1) | 131 (58.7) | 0.008 |
| Interstitial infiltration | 2 (7.1) | 56 (25.1) | 0.033 |
| Alveolar infiltration | 7 (25.0) | 8 (3.6) | <0.001 |
| Pleural effusion | 5 (17.9) | 13 (5.8) | 0.037 |
| Nodule/mass | 1 (3.6) | 4 (1.8) | 0.449 |
| **Treatment received** | 30 (93.8) | 216 (91.9) | 1.000 |
| **Medication** | N = 30 | N = 216 | |
| Amphotericin B initially | 14 (46.7) | 165 (76.4) | <0.001 |
| Itraconazole initially | 16 (53.3) | 51 (23.6) | |
| Median (IQR) days from diagnosis to treatment | 9.5 (5.0, 15.0) | 0 (0, 4.0) | <0.001 |
| Median (IQR) days from first symptom to treatment | 44.5 (21.8, 95.3) | 30.0 (20.0, 60.0) | 0.049 |
| 24-week mortality | 3/32 (9.4) | 36/235 (15.3) | 0.592 |

*Data are presented in number (%), mean ± standard deviation, and median (interquartile range) as appropriate

died from hospital-acquired infections, 4 from non-infectious causes, and 3 due to undetermined causes following referral to other hospitals.

### Comparisons of clinical characteristics of talaromycosis in patients with anti-interferon-γ autoantibody with other studies

There have been few reports of talaromycosis specifically in patients with anti-IFN-γ autoantibodies (AAbs). The details of studies conducted by Guo et al [7], and the present study are summarized in Table 3. A common finding across these studies is that the median age of patients was over 50 years, with the majority presenting with fever, weight loss, cough, and lymphadenopathy. The prevalence of skin lesions varied between studies, ranging from 26% to 69%.

## Discussion

Talaromycosis has been reported in HIV-uninfected patients with conditions such as anti-IFN-γ autoantibodies (AAbs), autoimmune diseases, organ transplantation, hematologic malignancies, and novel cancer therapies [4,14].

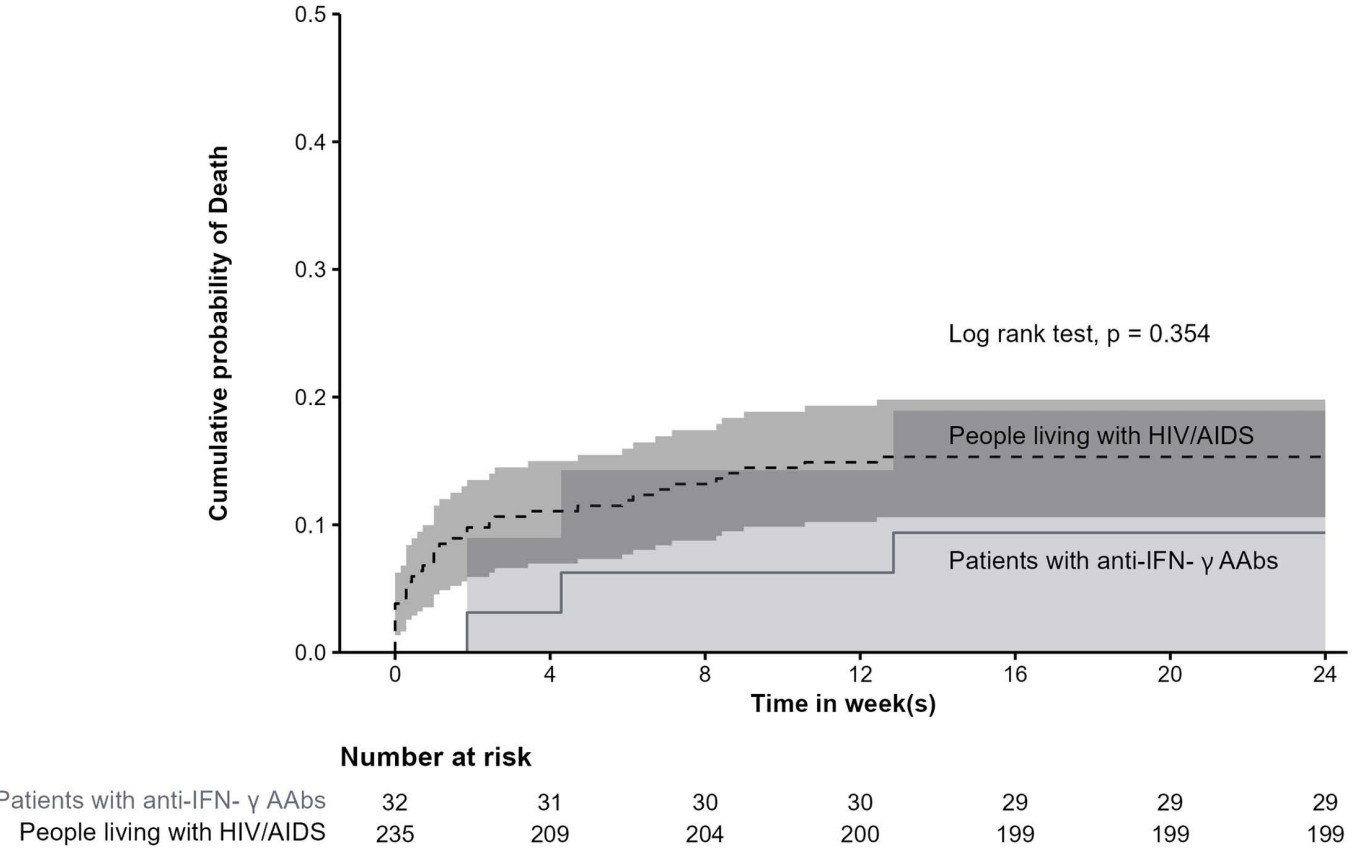

**Number at risk**

| | | | | | | | |
|---|---|---|---|---|---|---|---|
| Patients with anti-IFN- γ AAbs | 32 | 31 | 30 | 30 | 29 | 29 | 29 |
| People living with HIV/AIDS | 235 | 209 | 204 | 200 | 199 | 199 | 199 |

**Fig 2. Probability of death of patients with talaromycosis between patients with anti-IFN- γ AAbs and people living with HIV/AIDS.** Note: The shading represents the 95% confidence interval (CI). Two 95% CIs overlap.

Comparison of the clinical features of talaromycosis between patients with anti-IFN- γ-AAbs and PWHA has rarely been reported. In this study, the clinical features of talaromycosis were directly compared between these two groups, and some differences were found. Patients with anti-IFN- γ-AAbs were older (all were ≥ 40 years old) and had a higher proportion of underlying disease. In contrast to PWHA, patients with anti-IFN- γ-AAbs had previous infections before the diagnosis of talaromycosis, and the most common previous/current infections were non-tuberculous mycobacteria, tuberculosis, salmonellosis, and *Pneumocystis* pneumonia. These findings are consistent with previous reports of OI in patients with anti-IFN- γ-Aabs [5,6,15]. In addition, patients with anti-IFN- γ-AAbs had bone and joint infections and a lower proportion of fungemia. Since T cell-mediated immune responses play an important role in fighting *T. marneffei* infection, the advanced stage of HIV infection, which leads to a loss of CD4 + T cells and subsequently a loss of activation of cellular signaling pathways, results in an inability to control the infection [16], and fungemia was observed more frequently in PWHA [1–3]. Leukocytosis and higher platelet counts were found in patients with anti-IFN- γ AAbs, which could be due to the inflammatory cascade [17]. Abnormal CXR was found in 30% of patients with anti-IFN- γ AAbs. Since the route of transmission of talaromycosis is through inhalation of *T. marneffei* spores from the environment, the pulmonary symptoms and abnormal CXR are not unexpected [18]. The time from diagnosis of taralomycosis to initiation of treatment was longer in patients with anti-IFN- γ-AAbs, because patients who were not critically ill, were not hospitalized, and had a treatment appointment at the outpatient clinic that may not have occurred on the same day that cultures or pathology reports were available. This may explain why patients with anti-IFN- γ-AAbs were more likely to receive itraconazole initially than PWHA.

**Table 3. Comparisons of three studies of talaromycosis in patients with anti-interferon-γ autoantibody.**

| Demographic characteristics | This study | Guo et al [7] | Chen et al [9] |
|---|---|---|---|
| Years of study | January 2012-June 2021 | January 2013-June 2018 | January 2018-September 2020 |
| Study site | Chiang Mai, Thailand | China and Taiwan | Guangdong, China |
| Number of patients | 32 | 55 | 22 |
| Male | 19 (59.4) | 34 (58.6) | 14 (63.6) |
| Mean age (range) (years) | 56.9 (40.0, 75.0) | 54 (22, 77) | 52.0 (34.8, 58.0) |
| **Presence of underlying diseases that may compromised immune status** | | | NA |
| Diabetes mellitus | 2 (6) | 7 (13) | |
| Kidney disease | 4 (13) | 1 (2) | |
| Thalassemia | 1 (3) | 1 (2) | |
| Cirrhosis | 1 (3) | NA | |
| Malignancy | 1 (3) | NA | |
| **Clinical presentation** | | | |
| Median days from symptom onset to diagnosis (range) | 30 (21, 97) | 150 (28, 923) | 149.5 (57, 272) |
| **Signs/symptoms (%)** | | | |
| Fever | 21 (66) | 45 (82) | 12 (55) |
| Cough | 6 (19) | 39 (71) | 17 (77) |
| Dyspnea | 3 (9) | NA | 6 (27) |
| Weight loss/ wasting | 8 (25) | 42 (76) | 13 (59) |
| Lymphadenopathy | 20 (63) | 29 (53) | 18 (82) |
| Hepatomegaly | 7 (22) | 5 (9) | NA |
| Skin lesions | 22 (69) | 14 (26) | 10 (46) |
| Pleural effusion | 5 (18) | NA | 13 (59) |
| **Organ involvement (%)** | | | |
| Lung/pleura | 12 (43) | 55 (100) | NA |
| Lymph node | 20 (62) | 43 (78) | 18 (82) |
| Skin | 22 (69) | 14 (26) | 10 (46) |
| Bone and joint | 16 (50) | 13 (24) | 9 (41) |
| Liver | 7 (22) | 8 (15) | 2 (9) |
| Spleen | 0 | 5 (9) | 2 (9) |
| **Laboratory (median, IQR)** | | | |
| Hemoglobin (g/dL) | 8.6 (7.9, 10.0) | 9.2 (1.6, 12.0) | NA |
| White blood cells (x1000 cells/μL) | 18.1 (13.6, 23.8) | 16.1 (4.7, 37.7) | NA |
| Absolute neutrophil count (x1000 cells/μL) | 13.6 (9.7, 18.9) | 12.0 (2.7, 30.6) | NA |
| Mortality (%) | 3 (9) | 13 (24) | 3 (14) |

NA: not applicable

Itraconazole has a low minimum inhibitory concentration (MIC) against *T. marneffei* [19,20] and has been shown to be effective in the initial treatment of talaromycosis with mild disease [21,22]. The archived guidelines (through October 22, 2019) for the prevention and treatment of opportunistic infections in HIV-infected adults and adolescents from the Centers for Disease Control and Prevention, the National Institutes of Health, and the HIV Medicine Association of the Infectious Diseases Society of America suggest that initial treatment with itraconazole is the preferred therapy for patients with mild disease [23]. However, since the November 21, 2019, itraconazole is no longer recommended as induction therapy for

talaromycosis, as the results of IVAP trial in PWHA, which showed the superiority of amphotericin B over itraconazole in the induction phase, regardless of disease severity and faster fungal clearance in blood culture [2,23]. Whether similar results apply to other populations, including patients with anti-IFN- γ-AAbs, and whether faster fungal clearance may influence clinical outcomes in patients with anti-IFN- γ-AAbs remains to be determined. Not surprisingly, all but one patient with anti-IFN- γ-AAbs initially received itraconazole because they were not ill enough to be hospitalized in a hospital with limited admission capacity and were willing to receive oral therapy and be closely monitored in the outpatient clinic as they were able to return to work.

The mortality rate in PWHA (14.8%) was in the range of previous studies (12.6% - 33%) [2,7,10,24–26]. The rate in patients with anti-IFN- γ AAbs was 9%, which was lower than the rate reported by Chen et al. (14%) [9] and Guo et al. (22%) [7]. In this study, the mortality rate of talaromycosis in patients with anti-IFN- γ-AAbs was not statistically different from PWHA. Our earlier study showed that the mortality rate in HIV-uninfected patients was 20.7%. However, the exact proportion of patients with anti-IFN-γ-AAbs was not known [10].

The strength of this study is that it is one of the few studies comparing clinical features and outcomes of talaromycosis in patients with anti-IFN- γ-AAbs and PWHA from an area where *T. marneffei* is endemic and HIV infection is prevalent. This study raised awareness among physicians caring for these patients, especially in the endemic area.

However, there were several limitations. First, the sample size may not be large enough to detect differences between groups, if any. Second, as the nature of retrospective study, which was not designed for systematic data collection, there were data missing, i.e., not all patients had performed blood cultures, not all PWHA had baseline CD4 count, and chest radiograph. Third, patients with anti-IFN- γ-AAbs may have concurrent infections, so the timing of first symptom onset may not be accurate.

## Conclusions

The clinical features of talaromycosis in patients with anti-IFN- γ AAbs differed in some aspects from PWHA. Awareness of talaromycosis in patients with anti-IFN- γ AAbs, even in the absence of detection of *T. marneffei* in blood cultures, may shorten the time from initial symptoms to diagnosis and treatment.

## Author contributions

**Conceptualization:** Quanhathai Kaewpoowat, Romanee Chaiwarith.

**Data curation:** Kawisara Krasaewes, Narootchai Patanadamrongchai, Saowaluck yasri.

**Formal analysis:** Saowaluck yasri, Antika Wongthanee.

**Investigation:** Kawisara Krasaewes, Narootchai Patanadamrongchai, Jiraprapa Wipasa.

**Methodology:** Quanhathai Kaewpoowat, Romanee Chaiwarith.

**Supervision:** Romanee Chaiwarith.

**Validation:** Saowaluck yasri, Romanee Chaiwarith.

**Visualization:** Romanee Chaiwarith.

**Writing – original draft:** Kawisara Krasaewes, Narootchai Patanadamrongchai, Jiraprapa Wipasa, Saowaluck yasri.

**Writing – review & editing:** Quanhathai Kaewpoowat, Romanee Chaiwarith.

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
