## [Decision Letter · Decision Letter 0]

13 Sep 2023

Dear Dr Chaiwarith,

Thank you very much for submitting your manuscript "Talaromycosis: Differences between Patients with the Anti-interferon-γ autoantibody and people living with HIV/AIDS (PLWHA)" for consideration at PLOS Neglected Tropical Diseases. As with all papers reviewed by the journal, your manuscript was reviewed by members of the editorial board and by several independent reviewers. In light of the reviews (below this email), we would like to invite the resubmission of a significantly-revised version that takes into account the reviewers' comments. 

I have reviewed and agree with all suggestions by reviewers. In particular, I agree with the comment by reviewer #3 that the authors should include all 265 PLWHA over the study period, rather than randomly selected 64 patients as rationale for subgroup selection is unnecessary, unsubstantiated, and introduces bias. I also agree with the need to have the paper edited by an English speaking medical writer.

We cannot make any decision about publication until we have seen the revised manuscript and your response to the reviewers' comments. Your revised manuscript is also likely to be sent to reviewers for further evaluation.

Sincerely,

Thuy Le

Academic Editor

Marcio Rodrigues

Section Editor

I have reviewed and agree with all suggestions by reviewers. In particular, I agree with the comment by reviewer #3 that the authors should include all 265 PLWHA over the study period, rather than randomly selected 64 patients as rationale for subgroup selection is unnecessary, unsubstantiated, and introduces bias. I also agree with the need to have the paper edited by an English speaking medical writer.

Reviewer's Responses to Questions

**Key Review Criteria Required for Acceptance?**

**Methods**

-Are the objectives of the study clearly articulated with a clear testable hypothesis stated?

-Is the study design appropriate to address the stated objectives?

-Is the population clearly described and appropriate for the hypothesis being tested?

-Is the sample size sufficient to ensure adequate power to address the hypothesis being tested?

-Were correct statistical analysis used to support conclusions?

-Are there concerns about ethical or regulatory requirements being met?

Reviewer #1: - the objective is to compare clinical features and outcome between talaromycosis in patients with advanced HIV disease and an auto-immune disorder.

- the design of a retrospective cohort study is appropiate here

- the population is clearly described, although the authors do not describe clearly how patients with advanced HIV were both matched and randomly selected.

- no sample size calculation was included

- as far as I can judge, correct stats were used for an observational study like this

- No concerns about ethical and regulatory requirements

Reviewer #2: (No Response)

Reviewer #3: This is an important manuscript describing the differences between two groups of patients with talaromycosis in the endemic area of northern Thailand. It has been previously described that patients with autoimmune conditions are also at risk of talaromycosis, alongside the traditional at-risk group of PLWHA, however this is a comprehensive review of the characteristics of both populations, and one of the first in Northern Thailand.

Major Revisions

1) Methodology and selection of PLWHA – it is unclear from the methods section why 64 patients were selected from the underlying population of 265 PLWHA. Were they matched with the auto-immune group and what characteristics were used for matching? If they were not matched, what was the purpose of selecting 64 patients? This is a descriptive paper rather than an analytic paper, my suggestion here would be to include all 265 patients in the analysis rather than randomly selecting 64. A description of the entire PLWHA cohort during the study period would be of most value in the current paper.

2) Materials and methods - overall this section does not contain enough detail, particularly in justifying the choice of 64 patients randomly selected from the underlying PLWHA cohort

3) Discussion section - this section should emphasize what this manuscript adds to the overall literature on this topic. This rationale is not evident in the discussion section. Authors should emphasize in the early sentences of this section what this manuscript adds to the current literature. 

Minor Revisions

1) Grammer, syntax and typographical errors– This manuscript would benefit from the review of a medical writer or medial editor. While the errors in English syntax and grammar, as well as typographical errors are minor, however they are numerous throughout the manuscript. Correcting these errors would improve the readability of the manuscript and I believe are essential prior to publication

2) Table 1: Row headings need to include what the value in brackets refers to. Ie for previous/concurrent OIs, the values within parentheses presumably refer to column percentages.

3) Table 1: Median (IQR) days from symptom to diagnosis, for the autoimmune column there should be a comma between 14 and 75)

4) Fig 1: The authors note that there is not "statistically significant [difference]" between the two groups, however the Fig does seem to demonstrate the the KM curves are separate and do not cross. Is it correct that there are no differences in mortality between the two groups? Also note typographic error in the Mortality section of the results ("statistically significant...")

**Results**

-Does the analysis presented match the analysis plan?

-Are the results clearly and completely presented?

-Are the figures (Tables, Images) of sufficient quality for clarity?

Reviewer #1: - it is not clear whether all data were available for all patients and what was done with missing data (if any) - tha authors only mention missing data in line 191

- the tables and figure mostly present the data clearly and completely.

- in table 2 it is unclear what is meant with treatment and antifungals... is this treatment given on admission, or treatment given after T. marneffei infection was diagnosed? 

minor:

line 76-77: were abscesses and nodules caused by T. marneffei, or other pathogens or unknown? The umbilicated lesions are basically pathognomonic for talaromycosis, the abscesses / nodules could be many other things

line 93-103: not sure if this extensive narrative on 2 patients is needed for this manuscript, probably not.

Reviewer #2: (No Response)

Reviewer #3: (No Response)

**Conclusions**

-Are the conclusions supported by the data presented?

-Are the limitations of analysis clearly described?

-Do the authors discuss how these data can be helpful to advance our understanding of the topic under study?

-Is public health relevance addressed?

Reviewer #1: - the authors could consider to present the comparison with two other studies looking at HIV negative patients as a result rather than discussion

- otherwise no issues

Reviewer #2: (No Response)

Reviewer #3: (No Response)

**Editorial and Data Presentation Modifications?**

Reviewer #1: - I don't think it's appropriate to thank patients who didn't consent and doctors who weren't aware of the study at the time in line 203-4

Reviewer #2: (No Response)

Reviewer #3: (No Response)

**Summary and General Comments**

Reviewer #1: The authors describe the results of a retropspective cohort study in a single hospital in Thailand, comparing talaromycosis patients with advanced HIV disease and with adult-onset immunedeficiency caused by anti-IFN-gamma autoantibodies.

The introduction could do with some more information on burden of both talaromycosis and the described immunedeficiency. Some more clinical and epidemiological information on the immunedeficiency would also be helpful for the PLoS NTD readership - especially about presentation of other opportunistic infections. As talaromycosis appears to have a more pulmonary presentation in these patients, is this similar for TB and NTB mycobacteria? Does this disease mainly associate with lung infections? How does salmonellosis present, by BSI?

There are some minor language errors throughout the manuscript.

Reviewer #2: This article elucidates the disparities in clinical presentations of Talaromycosis among two distinct populations. Notably, there is a paucity of literature regarding the clinical manifestations of Talaromycosis in patients exhibiting anti-interferon-gamma autoantibod rendering this study innovative and possessing significant implications for clinical practice.

1. The anti-interferon-gamma autoantibody is not clearly explained in the background knowledge.

2. The detection method for anti-interferon-gamma autoantibody should be explained in detail.

3. Is it also possible for HIV/AIDS patients to have anti-interferon-gamma autoantibody? Should these patients be excluded?

4. A flowchart explaining the process of screening the thirty-two patients with anti IFN- γ AAb should be provided.

5. The expression of CD4+T lymphocytes is inaccurate.

6. The word 'Gou' should be 'Guo' on page 13, line 180.

7. This study shows a lower patient mortality rate than the literature generally, but the reasons have not been explained.

8. The paper presentation requires significant improvement. In its present form, it is hard to read. There are many typos, syntax and grammatical errors that it is very difficult to enumerate them all. For example:

1) Talaromycosis: Differences between Patients with the Anti-interferon-γ autoantibody and people living with HIV/AIDS (PLWHA).

2) This study aimed to compare characteristics, and mortality associated with talaromycosis with adult-onset immunodeficiency caused by the anti-interferon-gamma autoantibody (anti-IFN-γ AAb) to those of people living with HIV/AIDS (PLWHA)

3) Table 1: 30 (14. 75) -> 30 (14, 75)

4) Line 20-21: however, a cause of immunodeficiency was not fully investigated -> however, the cause of immunodeficiency was not fully investigated.

5) If a patient lost to follow up or was referred out to other hospital, the status of a patient was retrieved from the Thai national civil registration, if a patient died, the date of death was obtained. 

6) Line180: Gou et al (22%) -> Guo et al. (22%)

9. The methods description in your Abstract section is not clear. Please update it.

10. The following sentence should be added in Data Analysis section.

“Statistical significance was set as a P ≤ 0.05 for all analyses.”

11. The quantitative variables should be described with median rather than the mean.

12. “PLWHA” should be used when the term “people living with HIV/AIDS” appeared in second time.

13. The following description (Line 83-84) is inconsistent with the content of Table 2. Please check it carefully. 

“Abnormal chest radiograph was less common in patients with anti-IFN- γ AAbs.”

14. Some terms that appeared in your Results section cannot be found in any Tables and Figures. For example: leukocytosis, paravertebral abscess, subcutaneous abscess. Please check it carefully. 

15. Please unify the writing of digital that in your Results section and Tables.

16. The following descriptions (Line 93-103) in Treatment section are not the part of your study results. Please remove it and update this section.

“For the other 2 PLWHA, physicians missed the diagnosis of talaromycosis. The first patient presented with low grade fever, and cervical and supraclavicular lymphadenopathy. While the source of fever was investigating, the patient received itraconazole 200 mg daily for primary fungal prophylaxis and visited the clinic 1 week later with defervescence. The regimen was not changed. The second patient presented with fatigue and dyspnea. Trimethoprim-sulfamethoxazole along with prednisolone were started for Pneumocystis jirovecii pneumonia. Fluconazole 400 mg. weekly was started for primary fungal prophylaxis. The patient visited the clinic 3 weeks later with clinical improvement. Itraconazole for the first patient and fluconazole for the second patient were discontinued when achieving CD4 cell count of ≥200 cells/cu.mm. for 6 months. Both patients were survived at the time of conducting this study”.

Reviewer #3: (No Response)

PLOS authors have the option to publish the peer review history of their article (what does this mean? ). If published, this will include your full peer review and any attached files.

**Do you want your identity to be public for this peer review?** For information about this choice, including consent withdrawal, please see our Privacy Policy .

Reviewer #1: Yes: H Rogier van Doorn

Reviewer #2: No

Reviewer #3: No
---

## [Decision Letter · Decision Letter 1]

24 Jun 2024

Dear Dr Chaiwarith,

Thank you very much for submitting your manuscript "Clinical features of talaromycosis in people living with HIV/AIDS (PLWHA) and patients with anti-interferon-γ autoantibodies" for consideration at PLOS Neglected Tropical Diseases. As with all papers reviewed by the journal, your manuscript was reviewed by members of the editorial board and by several independent reviewers. All reviewers appreciated the attention to the requested revision. However, please address the remaining concerns and modify the manuscript according to the review recommendations. 

Sincerely,

Thuy Le

Academic Editor

Marcio Rodrigues

Section Editor

Reviewer's Responses to Questions

**Key Review Criteria Required for Acceptance?**

**Methods**

-Are the objectives of the study clearly articulated with a clear testable hypothesis stated?

-Is the study design appropriate to address the stated objectives?

-Is the population clearly described and appropriate for the hypothesis being tested?

-Is the sample size sufficient to ensure adequate power to address the hypothesis being tested?

-Were correct statistical analysis used to support conclusions?

-Are there concerns about ethical or regulatory requirements being met?

Reviewer #1: No further comments

Reviewer #2: The objectives of the study were articulated clearly

The study design was appropriate to address the stated objectives

The population were clearly described and appropriate for the hypothesis being tested

The sample size was a bit small

The statistical analysis was correctly used to support conclusions

There were no concerns about ethical or regulatory requirements being met.

Reviewer #3: The authors have corrected the major methodological challenge with the original submission, which was around selection of the HIV patients. This has now been rectified with inclusion of all 235 HIV patients. This vastly strengthens the conclusions of the paper, and I note some of the results have indeed changed with inclusion of the larger HIV cohort. However I note that Table 2 still only discusses 64 HIV patients - why is this?

**Results**

-Does the analysis presented match the analysis plan?

-Are the results clearly and completely presented?

-Are the figures (Tables, Images) of sufficient quality for clarity?

Reviewer #1: No further comments

Reviewer #2: 1.In Table 2, all individuals with anti-IFN- γ-AAbs used itraconazole. So, what prevents patients from choosing AMB

2.Does the mortality rate include the untreated population? How to determine the mortality rate of patients who have been transferred to other hospitals

3.Are the deaths mainly caused by talaromycosis or other comorbidities?

Reviewer #3: 1) Line 128, I don't understand what this sentence means " Ten PLWHA survived at the time this study was conducted." This seems like a high mortality rate among HIV patients if correct. 

2) Line 130 - need to specify that 102/216 refers to HIV patients (if this is the case)

3) Line 148 - could phrase this better ie "This study has shown that the clinical manifestations of talaromycosis varies depending on the immunocompromising condition of the host."

4) Line 234 - check this paragraph. Lists first limitation then jumps to third limitation. I also think the limitiation section in general could do with some further thought - i.e this is a retrospective studies which introduces bias, missing data, unmeasured confounders, etc

**Conclusions**

-Are the conclusions supported by the data presented?

-Are the limitations of analysis clearly described?

-Do the authors discuss how these data can be helpful to advance our understanding of the topic under study?

-Is public health relevance addressed?

Reviewer #1: No further comments

Reviewer #2: The conclusions are supported by the data presented

The limitations of analysis are partially described

The authors discuss how these data can be helpful to advance our understanding of the topic under study.

The data collected in this study provided Clinicians with a better understanding of some of the different features of talaromycosis in patients with anti-IFN- AAbs and PLWHA.

Reviewer #3: NA

**Editorial and Data Presentation Modifications?**

Reviewer #1: Line 127-8: Correct to: "In three PLWHA, physicians missed the diagnosis of talaromycosis but these three PLWHA received..."

Line 128-9: What do the authors mean with: "Ten PLWHA survived at the time this study was conducted." - seems a very low number out of a group of 235 patients, typo - or am I misunderstanding something?

Line 130: Does it require explanation that the recommended treatment of AmphoB is consistently not used in patients with anti IFN-gamma abs? Is this per guidelines? The proportion of PLWHA receiving amphoB is also very low. Any mortality differences observed between the amphoB and itra only groups?

Line 222 and further: should it just be "mortality", rather than "mortality rate" (5 times)

Reviewer #2: Minor Revision

Reviewer #3: This revision is written clearly and easy to understand. I did not pick up any major errors. Minor errors as follows:

1) line 114 - needs a full stop.

2) Table 2 and Table 3 - it is not clear what the values mean for some of the rows. Need to specify what the figures in brackets are referring to.

**Summary and General Comments**

Reviewer #1: The manuscript has improved with the added full PLWHA group, and after addressing other comments by 3 reviewers. I am happy to recommend publication, some minor discretionary suggestions still

Reviewer #2: 1. It is unique and valuable that this study describes the clinical differences of talaromycosis patients between anti IFN- γ AAbs and PLWHA. As a result of this study, it is possible to gain an understanding of the characteristics and prognosis of talaromycosis in two populations. The shortcoming is that the background has not yet clearly explained the differences in treatment and prognosis between the two groups. This makes the data on treatment and prognosis appear abrupt. Furthermore, no in-depth discussion or analysis of some meaningful differences was provided in the discussion section.

2. The three reviewers of the manuscript had raised a lot of detailed questions. However, the authors' reply was not very satisfactory, especially some key questions. For example, the reviewers mentioned the need for a clear flowchart, the background and detection details of anti-interferon-gamma autoantibody, the lack of detailed description of the methodology, and so on, which have not been substantially modified in the manuscript.

3. There are still some problems with the manuscript, such as "Demographic data, clinical characteristics, laboratory findings, treatment and treatment results were recorded" mentioned by the authors, so where does the information come from, how to obtain it, and how to ensure authenticity, reliability and standardization? More details need to be provided.

4. The authors mentioned a total of 38 deaths, so do you analyze the specific causes of their deaths?

5. With regard to the limitations, the author did not seriously evaluate its potential impact on this study and the direction of future research that needs to be improved. In addition, the second point of limitation is missing.

5. The discussion part is limited to superficial comparison and lack of conciseness. In addition, it is not clear what this study has added to the previous literature, what scientific significance it has, and what kind of impact it has on clinical practice.

Reviewer #3: NA

PLOS authors have the option to publish the peer review history of their article (what does this mean? ). If published, this will include your full peer review and any attached files.

**Do you want your identity to be public for this peer review?** For information about this choice, including consent withdrawal, please see our Privacy Policy .

Reviewer #1: Yes: H Rogier van Doorn

Reviewer #2: No

Reviewer #3: No

Figure Files:

Data Requirements:

Reproducibility:

References

---

## [Editor Report · Decision Letter 2]

16 Oct 2024

Dear Dr Chaiwarith,

Thank you very much for submitting your manuscript "Clinical features of talaromycosis in people living with HIV/AIDS (PLWHA) and patients with anti-interferon-γ autoantibodies" for consideration at PLOS Neglected Tropical Diseases. As with all papers reviewed by the journal, your manuscript was reviewed by members of the editorial board and by several independent reviewers. The reviewers appreciated the attention to an important topic. Based on the reviews, we are likely to accept this manuscript for publication, providing that you modify the manuscript according to the review recommendations. 

The Editors have made some edits throughout the paper to improve the grammar and clarity of the paper and have made specific requests of revisions before the paper can be considered for publication in PLOS NTD. Please make the requested revisions and keep all tracked changes for ease of review. The suggested revisions are specified in the edited manuscript attached.

Sincerely,

Thuy Le

Academic Editor

Marcio Rodrigues

Section Editor

The Editors have made some edits throughout the paper to improve the grammar and clarity of the paper and have made specific requests of revisions before the paper can be considered for publication in PLOS NTD. Please make the requested revisions and keep all tracked changes for ease of review. The suggested revisions are specified in the edited manuscript attached.

Figure Files:

Data Requirements:

Reproducibility:

References

---

## [Editor Report · Decision Letter 3]

17 Mar 2025

Dear Dr Chaiwarith,

We are pleased to inform you that your manuscript 'Clinical features of talaromycosis in people living with HIV/AIDS (PWHA) and patients with anti-interferon-γ autoantibodies' has been provisionally accepted for publication in PLOS Neglected Tropical Diseases.

Best regards,

Marcio L Rodrigues

Section Editor

Marcio Rodrigues

Section Editor

Shaden Kamhawi

co-Editor-in-Chief

Paul Brindley

co-Editor-in-Chief

---

## [Editor Report · Acceptance letter]

Dear Dr Chaiwarith,

We are delighted to inform you that your manuscript, "Clinical features of talaromycosis in people living with HIV/AIDS (PWHA) and patients with anti-interferon-γ autoantibodies," has been formally accepted for publication in PLOS Neglected Tropical Diseases.

Best regards,

Shaden Kamhawi

co-Editor-in-Chief

Paul Brindley

co-Editor-in-Chief
